# Prevalence and determinants of anemia among pregnant women in East Africa; A multi-level analysis of recent Demographic and Health Surveys

**Alemneh Mekuriaw Liyew**[1]*, **Getayeneh Antehunegn Tesema**[1], **Tesfa Sewunet Alamneh**[1], **Misganaw Gebrie Worku**[2], **Achamyeleh Birhanu Teshale**[1], **Adugnaw Zeleke Alem**[1], **Zemenu Tadesse Tessema**[1], **Yigizie Yeshaw**[1,3]

1 Department of Epidemiology and Biostatistics, Institute of Public Health, College of Medicine and Health Sciences and Comprehensive Specialized Hospital, University of Gondar, Gondar, Ethiopia, 2 Department of Human Anatomy, College of Medicine and Health Sciences and Comprehensive Specialized Hospital, University of Gondar, Gondar, Ethiopia, 3 Department of Human Physiology, College of Medicine and Health Sciences and Comprehensive Specialized Hospital, University of Gondar, Gondar, Ethiopia

* alemnehmekuriawliyew@gmail.com

**Data Availability Statement:** The data used in this study are third party data from DHS (http://www.dhsprogram.com) and can be accessed

## Abstract

### Introduction

Anemia during pregnancy is a public health problem that leads to different life-threatening complications and poor pregnancy outcomes. So far, the evidence is scarce on pooled prevalence and determinants of anemia during pregnancy in East Africa for integrated intervention. Therefore, this study aimed to assess the prevalence and determinants of anemia among pregnant women in eastern Africa using recent Demographic and Health Surveys.

### Method

Secondary data analysis was conducted using data from recent Demographic and Health Survey datasets from 10 East African countries. A total of 8583 (weighted sample) pregnant women were included in the analysis. The multi-level mixed-effects generalized linear model (Poisson regression with robust error variance) was fitted to identify determinants of anemia. Finally, the adjusted prevalence ratio (aPR) with 95% CI and random effects for the multilevel generalized linear mixed-effects model was reported.

### Results

In this study, the overall prevalence of anemia among pregnant women was 41.82% (95% CI: 40.78, 42.87) with a large difference between specific countries which ranged from 23.36% in Rwanda to 57.10% in Tanzania. In the multi-level analysis, teenage pregnant women (aPR = 1.22;95%CI:1.02, 1.40), unmarried women (aPR = 1.14; 95% CI;1.02,1.28), pregnant women who had unimproved toilet facility (aPR = 1.17;95%CI:1.06,1.27), and those women from countries with high illiteracy level (aPR = 1.12;95%CI; 1.07,1.18) had a higher prevalence of anemia during pregnancy.

following the protocol outlined in the Methods section.

**Funding:** The authors received no specific funding for this work.

**Competing interests:** The authors have declared that no competing interests exist.

**Abbreviations:** ANC, Antenatal Care; aPR, Adjusted Prevalence Ratio; CSA, Central Statistical Agency; DHS, Demographic and Health Survey; EAs, Enumeration Areas; WHO, World Health Organization.

## Conclusion

Anemia is still a public health problem in East Africa. Therefore, enabling the households to have improved toilet facilities by strengthening the existing health extension program, reducing teenage pregnancy, and improving the community literacy level is vital to reduce the prevalence of anemia during pregnancy in East Africa.

## Introduction

Anemia during pregnancy refers to a hemoglobin concentration of less than 11 g/dL [1]. It is the most common hematologic disorder which affects the normal functioning of the organ system by creating a scarcity of oxygen that reaches different tissues and organs through blood circulation [2]. Although anemia can occur among any human population, pregnant women and young children are common victims of this hematologic abnormality. The hemoglobin deprivation due to anemia during pregnancy has serious maternal and fetal complications, which could even lead to maternal mortality [3]. The evidence shows that anemia contributes to 20% of deaths among pregnant women [4]. The main causes of anemia during pregnancy are nutritional deficiencies (iron, vitamin B12, folate), parasitic infections, (hookworm and malariae.t.c) [5], and acute blood loss [6].

Globally, about 32.4 million pregnant women were anemic where Southeast Asia and Africa share about 48.7% and 46.3% of the anemia burden respectively [7]. The highest rate of anemia during pregnancy is hosted in the Sub-Saharan region where 17.2 million pregnant women were reported to be anemic [8]. The prevalence of anemia among pregnant women in East African countries ranges from 20% in Rwanda [9] to 32.5% in Uganda [10]. Besides, it varies across different countries in the other part of the world [11–16].

Therefore, such a higher anemia burden during pregnancy is a major public health issue since it puts the affected women at higher risk of numerous complications to fetus and women herself during and after pregnancy [17–19]. Its negative health consequence continues through the period of infancy with long-lasting poor infant outcomes unless the disorder is corrected early [20,21]. Besides, the specific common bad consequences of anemia are intrauterine growth retardation, preterm delivery, low birth weight, and fetal death [22]. It is also globally considered as an indicator of different adverse health and socioeconomic consequences since anemia impairs physical health, cognitive development, productivity, and reflects the poor economic development of a country [7,23].

Different national nutrition programs and micronutrient deficiency prevention and control strategies have been implemented to reduce anemia among pregnant women [24,25]. Despite the various efforts made maternal anemia is still a major public health concern [26].

In the previous studies, wealth index [27,28] maternal education [28–30], maternal age [31], parity [28,31,32], place of residence [30,33], maternal occupation [30], history of terminated pregnancy [28,33,34], iron intake during pregnancy [32,34], unimproved source of water [27] and marital status [35] were factors associated with anemia during pregnancy.

Though there were pieces of evidence regarding the effect of anemia during pregnancy in East Africa in the previous literature [9,15,27,35,36], none of these indicated the overall burden of anemia among pregnant women since they are studies in specific countries. Whereas others [37–40]are sub-country studies. Since in recent times there is a need to integrate East Africa in the health aspect to realize universal health coverage as part of sustainable development goals [41], the findings in the current study could have positive implications in this regard. Besides,

the benefit of this study from the one conducted on reproductive-age women [42] in this region is, it focuses on anemia burden among pregnant women. Since pregnancy is a highly oxygen-demanding period due to physiologic changes [3], the effect of anemia on pregnant women is superior to other reproductive-age women.

Furthermore, East African countries continued to be the hotspot areas of anemia. Therefore, to reduce the burden of anemia among pregnant women, it is vital to investigate the pooled prevalence and its determinants among pregnant women at the East African level. conducting pooled analysis using the nationally representative DHS data of East African countries is vital for understanding common determinants across countries. To reduce anemia incidence, the intervention of multi-sectoral organizations and international stakeholders to the common factors across countries is needed.

Besides, the findings of this study could help to design evidence-based public health decisions for reducing the incidence of anemia among pregnant women in East Africa, and consequently improve pregnancy outcomes. Moreover, this study was a pooled analysis that could increase the study power to permit a full examination of effect modification within the data and can reduce the measurement errors and bias arising when studies are combined that used heterogeneous designs and data collection methods.

## Method

### Study design and setting

This study used Demographic and Health Survey (DHS) data which were collected using a cross-sectional study design. Demographic and Health Surveys (DHS) are comparable nationally representative household surveys that have been conducted in more than 85 countries worldwide since 1984. The DHS collects a wide range of objective and self-reported data with a strong focus on indicators of fertility, reproductive health, maternal and child health, mortality, nutrition, and self-reported health behaviors among adults. Key advantages of the DHS include high response rates, national coverage, and high-quality interviewer training, standardized data collection procedures across countries, and consistent content over time. Data from DHS facilitate epidemiological research focused on monitoring prevalence, trends, and inequalities. It drew nationally representative samples for the country's population. A detailed description of the nature of demographic and health survey datasets was published elsewhere [43]. Therefore, the current study was based on demographic and health surveys which were conducted between 2008/09 and 2018/2019 in East African countries.

### Data source and measurements

The data for this study were drawn from recent nationally representative DHS data conducted in 10 (Burundi, Ethiopia, Madagascar, Malawi, Mozambique, Rwanda, Tanzania, Uganda, Zambia, Zimbabwe) countries in East Africa. Sudan and Eritrea had no recently conducted DHS data. The other East African countries such as Comoros,and Kenya, had no recorded data on the anemia status of women in their demographic and health survey dataset. The DHS surveys are routinely collected every five years across low- and middle-income countries using structured methodologies pretested and validated tools. The DHS surveys follow the same standard sampling procedure, questionnaires, data collection, and coding which is internationally led by the DHS program [44]. This makes multi-country analysis simple and reasonable.

To assure national representativeness, the DHS survey employs a stratified two-stage sampling technique in each country. In the first stage, enumeration areas (EAs) that represent the entire country were randomly selected from the sampling frame (i.e. developed from the

**Table 1. The study participants by country and respective year of the survey.**

| Country | Year of survey | Frequency (n) | Percent (%) |
|---|---|---|---|
| Burundi | 2016 | 694 | 8.08 |
| Ethiopia | 2016 | 1,135 | 13.23 |
| Madagascar | 2008/09 | 714 | 8.32 |
| Malawi | 2015/2016 | 639 | 7.44 |
| Mozambique | 2018 | 1,516 | 17.66 |
| Rwanda | 2014 | 491 | 5.72 |
| Tanzania | 2015 | 1,118 | 13.03 |
| Uganda | 2016 | 614 | 7.15 |
| Zambia | 2018/2019 | 1,083 | 12.62 |
| Zimbabwe | 2015 | 579 | 6.74 |
| Total | - | 8583 | 100 |

available latest national census). The second stage is the systematic sampling of households listed in each cluster or EA and interviews are conducted in selected households with target populations (women aged 15–49 and men aged 15–64). In this study, women aged 15–49 years who were pregnant during the survey period were included. Those pregnant women with no measured hemoglobin were excluded from the study. Therefore, the total sample size from the pooled (appended) data analyzed in this study was 8583, and the total number of pregnant women from each country was ranged from 491 in Rwanda to 1516 in Mozambique (Table 1).

## Ethical approval and consent to participate

Ethical approval for this study was not required since this study used existing public domain survey data sets, which are freely available online at www.measuredhs.com website with all identifier information removed. But to access and use the data we obtained permission and approval from MeasureDHS through the online request.

## Dependent variable

The dependent variable for this study was the anemia status. Pregnant women with the altitude-adjusted hemoglobin value <11 g/dL were classified as anemic otherwise nonanemic. Anemia is recorded as a categorical variable with nonanemic, mild, moderate, and severe categories in DHS datasets for each country. For this study, we recategorized that mild, moderate, and severe anemia as anemic (coded as "1") and non-anemic (coded as "0") to fit the multilevel binary logistic regression model. This was done because there were very small numbers of cases in the severe and moderate anemia category.

## Independent variables

From the most recent DHS datasets, educational status of the mother (no formal education, primary, secondary, higher education), maternal age (15–19, 20–24, 25–29, 30–34, 35–39, 40–49), maternal occupation (working, not working), parity (nulipara, primipara, multipara, grand multipara), wealth status (poorest, poorer, middle, richer and richest), history of a terminated pregnancy (yes, no), health insurance (yes, no), perception of distance from the health facility (big problem, not a big problem), iron supplementation (yes, no), media exposure (yes, no), source of drinking water (improved, not improved), type of toilet facility (improved, not improved), sex of household head (male, female) and marital status (unmarried, married) were considered as individual-level variables.

Whereas, place of residence (urban, rural), community poverty level (low, high), community illiteracy level (low, high), community health insurance (low, high) were considered as the community-level factors.

The aggregate community level explanatory variables (community poverty level, community illiteracy level, community health insurance) were computed by aggregating individual-level characteristics at the country level. They were dichotomized as high or low based on the distribution of the proportion values computed for each community after checking the distribution by using the histogram. If the aggregate variable was normally distributed mean value and if not, normally distributed median value was used as a cut-off point for the categorization.

## Statistical analysis

Data Extraction, recoding, and both descriptive and analytical analysis were carried out using STATA version 14 software. Weighting was done to restore the representativeness of the sample so that the total sample looks like the country's actual population. Descriptive analysis was conducted using cross-tabulation. As a result, frequencies and percentages were reported. The multilevel analysis was fitted due to the hierarchical nature of the demographic health survey data. In this study, the multilevel mixed-effects generalized linear model (using Poisson regression with robust error variance) was employed since the prevalence of anemia was high and the dependent variable was binary. Besides, the Intraclass Correlation Coefficient (ICC), and Proportional Change in Variance (PCV), were conducted to assess the variability across the country. Bi variable analysis was first done to select variables for multivariable analysis and variables with p-value <0.20 in the bivariable analysis were eligible for the multivariable analysis. After the candidate variables were selected in the bivariable analysis four models were fitted; the null model (with no predictors), model II (adjusted for individual-level variables only), model III (adjusted for community-level variables only), and model IV (model adjustment for both individual and community-level variables simultaneously) were fitted. the deviance was used for model comparison. Finally adjusted prevalence ratio (aPR) with a 95% confidence interval (CI) was reported for the best-fitted model.

## Results

### Sociodemographic characteristics of study participants

A total of 8583 pregnant women were included in this study. Of this, 24.20% were uneducated and 45.12% had no occupation. About 46% of participants perceived distance to the health facility as a big problem. Nearly half (49.85%) of the pregnant women used unimproved sources of drinking water and 35.62% had unimproved toilet facilities. The majority (68.91%) of the participants were married. Regarding the community-level characteristics, the majority of participants (77.83%) were from rural communities and nearly half (49.67%) of them were from countries with high illiteracy levels. Finally, about 52% and 49% of participants were from countries with high poverty levels and low health insurance coverage respectively (Table 2).

### Prevalence of anemia by country

The prevalence of anemia among pregnant women in eastern Africa was 41.82 (95%CI: 40.78, 42.87) with a large difference between specific countries which ranges from 23.36% in Rwanda to 57.10% in Tanzania (Fig 1).

**Table 2. The sociodemographic characteristics of study participants, East Africa.**

| Variables | Weighted Frequency(n) | Percent(%) |
|---|---|---|
| **Individual-level variables** | | |
| Maternal education | | |
| No formal education | 2,077 | 24.20 |
| Primary education | 4,463 | 52.00 |
| Secondary education | 1,803 | 21.00 |
| Higher education | 238 | 2.78 |
| Maternal age | | |
| 15–19 | 1,499 | 17.47 |
| 20–24 | 2,349 | 27.38 |
| 25–29 | 1,965 | 22.89 |
| 30–34 | 1,492 | 17.39 |
| 35–39 | 893 | 10.40 |
| 40–49 | 383 | 4.47 |
| Maternal occupation | | |
| Working | 4,709 | 55.88 |
| Not working | 3,873 | 45.12 |
| Parity | | |
| Nulipara | 1,990 | 23.19 |
| Primiparous | 1,763 | 20.54 |
| Multiparous | 3,314 | 38.62 |
| Grand multiparous | 1,515 | 17.65 |
| Wealth status | | |
| Poorest | 1,993 | 23.23 |
| Poor | 1,920 | 22.38 |
| Middle | 1,608 | 18.74 |
| Rich | 1,650 | 19.23 |
| Richest | 1,409 | 16.42 |
| History of a terminated pregnancy | | |
| Yes | 1,162 | 13.55 |
| No | 7,419 | 86.45 |
| Covered by health insurance | | |
| No | 7,681 | 89.51 |
| Yes | 900 | 10.49 |
| Distance from the health facility | | |
| Big problem | 3,948 | 46.00 |
| Not a big problem | 4,634 | 54.00 |
| Iron supplementation | | |
| Yes | 6,959 | 81.09 |
| No | 1,623 | 18.91 |
| Media exposure | | |
| Yes | 5,528 | 64.41 |
| No | 3,054 | 35.59 |
| Type of source of drinking water | | |
| Improved | 4,304 | 50.15 |
| Not improved | 4,278 | 49.85 |
| Type of toilet facility | | |
| Improved | 5,525 | 64.38 |

(*Continued*)

**Table 2.** (Continued)

| Variables | Weighted Frequency(n) | Percent(%) |
|---|---|---|
| Not improved | 3,057 | 35.62 |
| Sex of household head | | |
| Male | 6,847 | 79.78 |
| Female | 1,735 | 20.22 |
| Marital status | | |
| Unmarried | 2668 | 31.09 |
| Married | 5,914 | 68.91 |
| **Community-level variables** | | |
| Residence | | |
| Urban | 1,902 | 22.17 |
| Rural | 6,679 | 77.83 |
| Community poverty level | | |
| Low | 4,146 | 48.31 |
| High | 4,436 | 51.69 |
| Communityilliteracy | | |
| Low | 4,319 | 50.33 |
| High | 4,263 | 49.67 |
| Community health insurance | | |
| Low | 4,205 | 49.00 |
| High | 4,376 | 51.00 |

## Random effect and model comparison

Table 3 revealed the random effect or country-level variation of anemia and model comparison. Thus, in the null model, the variance component analysis was conducted to decompose the total variance of anemia. The country was used as a level two variable. Therefore, the country level variance which indicates the total variance of anemia that can be attributed to the context of the country in which the women were living was estimated. The applicability of multilevel analysis was justified by the significance of the country level variance [country variance = 0.30; standard error (SE) = 0.17; P-value = 0.001], indicating the existence of statistically significant differences between countries regarding anemia among pregnant women. This was further supported by the ICC in the null model which showed that about 8.50% of the variation of anemia among pregnant women was attributed to the difference at country-level factors. Besides, the final model(model IV) indicates that about 20% of the variation of anemia among pregnant women was attributable to both the individual level and country-level factors. Regarding model comparison, we used deviance to assess model fitness. Consequently, the model with the lowest deviance value (Model IV) was found to be the best-fitted model (Table 3).

## Determinants of anemia among pregnant women

As presented in Table 4, where both the individual and country-level factors were included simultaneously; maternal age, marital status, and type of toilet facility from individual-level factors and country illiteracy level from the aggregate country-level factors were significantly associated with anemia.

After controlling for other individual and community level factors, pregnant women who had unimproved toilet facility had 17% [adjusted prevalence ratio(aPR = 1.17; 95% CI: 1.06,

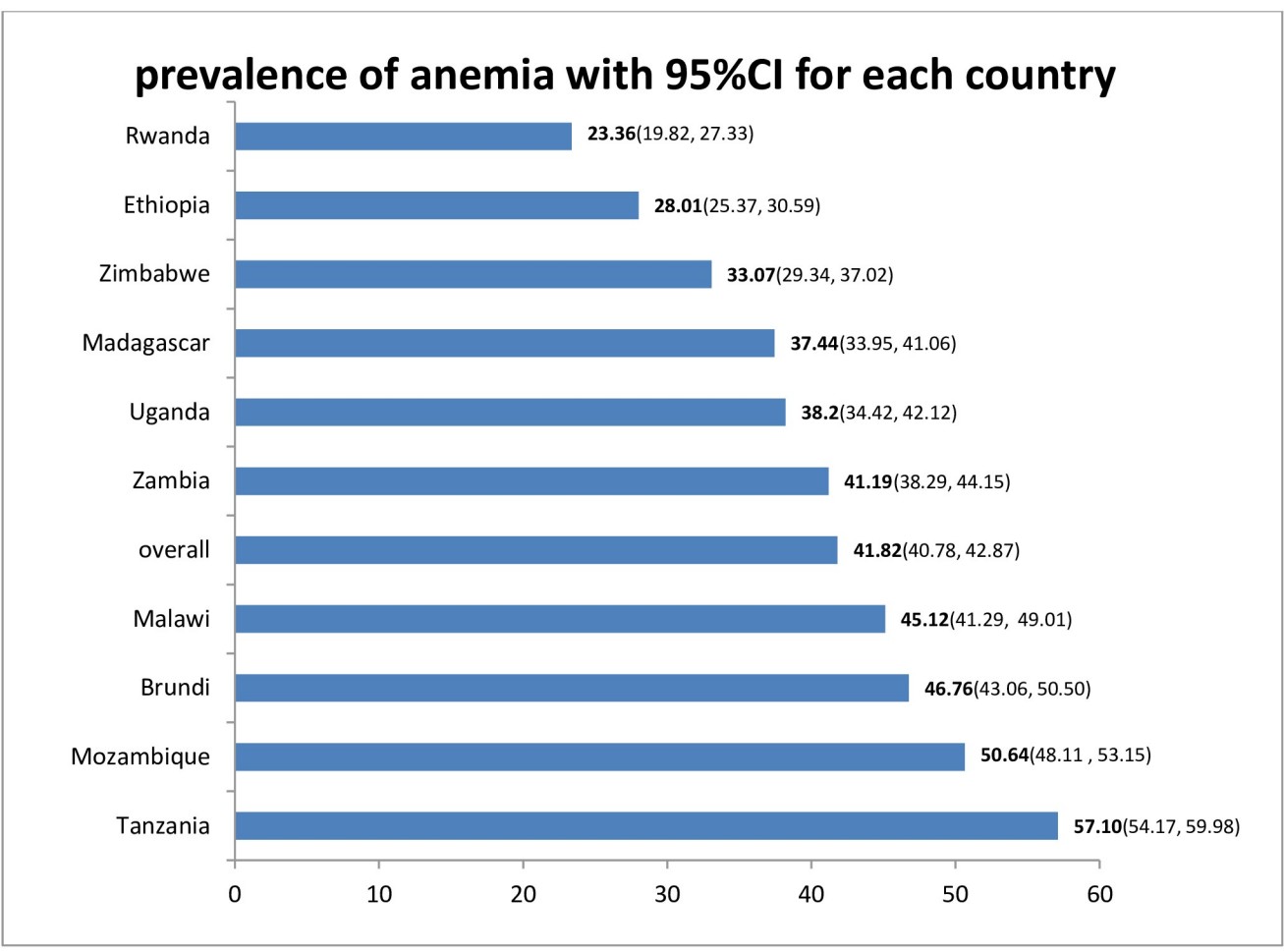

**Fig 1. Prevalence of anemia among pregnant women in East Africa, 2020.**

1.27) higher prevalence of anemia as compared to those who had improved toilet facility. Regarding maternal marital status, unmarried women had a 14% (aPR = 1.14; 95% CI;1.02, 1.28) higher prevalence of anemia as compared to married women.

The prevalence of anemia was increased by 22% (aPR = 1.22; 95%CI:1.02, 1.40) among teenagers as compared to women aged 40–49 years. Furthermore, pregnant women from the country with the higher illiteracy level had a 12% (aPR = 1.12; 95%CI; 1.07, 1.18) higher

**Table 3. Random effect and model comparison.**

| Parameter | Model I | Model II | Model III | Model IV |
|---|---|---|---|---|
| Community (country) variance(SE) | 0.30(0.17)* | 0.26(0.15)* | 0.27(0.16)* | 0.24(0.15)* |
| ICC (%) | 8.50(6.8,20.10) | 7.00(5.4,18.10) | 7.60(3.7,16.02) | 6.00(1.2,10.05) |
| PCV (%) | Reference | 13.33 | 10.00 | 20 |
| **Model fitness** | | | | |
| Deviance | 11,427.36 | 11,303.42 | 11,400.56 | 11,284.64 |

ICC: Intracluster correlation coefficient; PCV: Proportional change in variance;

*p-value<0.01.

**Table 4. Determinants of anemia among pregnant women in East Africa.**

| Variables | Model I(null) aPR 95%CI | Model II aPR 95%CI | Model III aPR 95%CI | Model IV aPR 95%CI |
|---|---|---|---|---|
| Maternal education | | | | |
| No formal education | - | 1.24(0.95,1.61) | - | 1.20(0.92,1.57) |
| Primary education | - | 1.19(0.94,1.50) | - | 1.19(0.95,1.50) |
| Secondary education | - | 1.15(0.90,1.47) | - | 1.15(0.90,1.47) |
| Higher education | - | 1.00 | - | 1.00 |
| Maternal age | | | | |
| 15–19 | | 1.22(1.02, 1.40) | | 1.22(1.02, 1.40)** |
| 20–24 | - | 0.89(0.82,1.20) | - | 0.88(0.81,1.26) |
| 25–29 | - | 0.87(0.75,1.00) | - | 0.86(0.75,1.00) |
| 30–34 | - | 0.85(0.69,1.05) | - | 0.84(0.69,1.05) |
| 35–39 | | 0.86(0.70,1.05) | | 0.85(0.71,1.05) |
| 40–49 | | 1.00 | | 1.00 |
| Maternal occupation | | | | |
| Working | - | 1.00 | - | 1.00 |
| Not working | - | 1.02(0.93,1.11) | - | 1.02(0.93,1.11) |
| Parity | | | | |
| Nulipara | - | 1.00 | - | 1.00 |
| Primiparous | - | 0.99(0.90,1.08) | - | 0.98 (0.90,1.08) |
| Multiparous | - | 0.96(0.79,1.17) | - | 0.96(0.81,1.22) |
| Grand multiparous | | 1.08(0.81,1.43) | | 1.03(0.27,1.34) |
| Wealth status | | | | |
| Poorest | - | 1.05(0.83,1.33) | - | 1.07(0.82,1.39) |
| Poorer | - | 1.04(0.87,1.24) | | 1.06(0.86,1.30 |
| Middle | - | 0.97(0.83,1.13) | - | 0.98(0.84,1.15) |
| Richer | - | 1.11(0.94,1.31) | | 1.12(0.95,1.32) |
| Richest | - | 1.00 | | 1.00 |
| History of terminated pregnancy | | | | |
| Yes | - | 1.00 | - | 1.00 |
| No | - | 1.02(0.95,1.09) | - | 1.01(0.95,1.09) |
| Covered by health insurance | | | - | |
| Yes | - | 1.00 | - | 1.00 |
| No | - | 1.01(0.88,1.15) | - | 1.01(0.89,1.15) |
| Distance from the health facility | | | | |
| Big problem | - | 0.98(0.91,1.04) | - | 0.97(0.92,1.03) |
| Not big problem | - | 1.00 | - | 1.00 |
| Iron supplementation | | | | |
| Yes | - | 1.00 | - | 1.00 |
| No | - | 1.03(0.94,1.12) | - | 1.02(0.94,1.12) |
| Media exposure | | | | |
| Yes | - | 1.00 | - | 1.00 |
| No | - | 0.98(0.88,1.09) | - | 0.98(0.88,1.09) |
| Source of drinking water | | | | |
| Improved | - | 1.00 | - | 1.00 |
| Not improved | - | 0.98(0.92,1.05) | - | 0.96(0.90,1.05) |
| Type of toilet facility | | | | |
| Improved | - | 1.00 | - | 1.00 |
| Not improved | - | 1.16(1.06,1.28) | - | 1.17(1.06,1.27)** |

(*Continued*)

**Table 4.** (Continued)

| Variables | Model I(null) aPR 95%CI | Model II aPR 95%CI | Model III aPR 95%CI | Model IV aPR 95%CI |
|---|---|---|---|---|
| Sex of household head | | | - | |
| Male | | 1.00 | - | 1.00 |
| Female | | 1.02(0.91,1.13) | - | 1.01(0.91,1.13) |
| Marital status | | | - | |
| Married | - | 1.00 | - | 1.00 |
| Unmarried | - | 1.13(1.01,1.27) | - | 1.14(1.02, 1.28)* |
| Residence | | | | |
| Urban | - | - | 1.00 | 1.00 |
| Rural | - | - | 1.04(0.98,1.11) | 0.97(0.91,1.04) |
| Community poverty level | | | | |
| Low | - | - | 1.00 | 1.00 |
| High | - | - | 1.01(0.96,1.06) | 0.99(0.94,1.04) |
| Community illiteracy level | | | | |
| Low | - | - | 1.00 | 1.00 |
| High | - | - | 1.14(1.06,1.18) | 1.12(1.07,1.18)*** |
| Community health insurance | | | | |
| Low | - | - | 0.97(0.89,1.06) | 0.98(0.89,1.07) |
| High | - | - | 1.00 | 1.00 |

**Note**: aPR: Adjusted Prevalence Ratio; CI: Confidence Interval;

* = P<0.05

** = P<0.01

*** = P<0.001.

prevalence of anemia as compared to those from countries with low community illiteracy levels (Table 4).

## Discussion

Anemia during pregnancy is related to increased maternal and child mortality and morbidity in low-income countries [14,45]. Thus, this study assessed the prevalence and determinants that affect anemia during pregnancy in East Africa. This study revealed that 41.82% (95% CI: 40.78, 42.87) of pregnant women were anemic, which indicates that anemia among pregnant women is a major public health problem in East Africa [3]. This finding was consistent with a study in Nigeria [11]. The prevalence in this study is higher than the studies conducted in Saudi Arabia [12], Ethiopia [36], and Uganda [38]. However, this finding is lower than the studies done in Mali [13], India [29], Sudan [15], and Pakistan [16]. Such geographical variations of anemia across the countries might be attributable to the difference in food preferences and cultural beliefs about dietary consumption during pregnancy [46], the occurrence of communicable diseases [47], and the difference in the availability of healthcare facilities. Besides, the higher prevalence of anemia in the current study could be attributed to the recent resurgences of malaria in East Africa due to climate change [48].

According to the final model, both individual-level and community-level factors account for 20% of the variation in anemia prevalence among pregnant women in East Africa. This study revealed that pregnant women who had unimproved toilet facilities had a higher prevalence of anemia as compared to those who had improved toilet facilities. This is in line with a study conducted using data from the most recent demographic and health surveys from 47

countries [49]. This might be because lack of clean water and unimproved toilet facilities would increase the occurrence of soil-transmitted infections [50] which might, in turn, lead to anemia [51]. Besides, the evidence indicates that increased hookworm infection intensity among pregnant women is related to lower blood hemoglobin levels in pregnant women in economically poor countries [52].

Similarly, this study highlighted that pregnant teenagers had a higher prevalence of anemia as compared to older women. This finding is consistent with studies done in Uganda [27], India [31], and Saudi Arabia [12]. Different pieces of evidence showed that early marriage is associated with low economic status, a dropout from education, risk of sexually transmitted infections (like HIV/AIDS), higher rates of several poor social and physical outcomes, complications of pregnancy (including anemia), and high rate of divorce [53–55]. Thus, the cumulative effect of such conditions might put teenage pregnancy at a higher risk of anemia.

The current study also indicated that the prevalence of anemia among unmarried women was higher as compared to married women. This result is supported by the finding in Rwanda [35]. The empirical evidence in different publications indicated that unmarried pregnant women are subjected to higher morbidity rates, poor emotional well-being [56,57] stress, and depression. They are less likely to report being happier and healthier as compared to married women [58]. Therefore, these may aggravate the already existing physiologic stress during pregnancy which might result in the suppression of the production of RBCs in the bone marrow. Furthermore, pregnancy in unmarried women is most likely unintended. Though daily iron supplementation, is a proven public health intervention for pregnant women, a study showed that a woman with unintended pregnancy has low compliance to iron supplementation (i.e iron deficiency is the most common cause of anemia) due to social stigma [59].

Of the community(country) level factors, community illiteracy level is significantly associated with anemia during pregnancy. Accordingly, the prevalence of anemia is high among women from countries with high illiteracy as compared to those from the country with low illiteracy levels. This finding was concordant with the findings from other publications [60,61]. This might be due to illiterate women may be economically unstable [62] and they may fail to achieve the nutritional requirement (foods rich in vitamin A, iron, and folic acid e.t.c) during pregnancy which may result in anemia. There is also a piece of evidence that these illiterate women did not go for antenatal care (ANC) service [63]. Unless they are supplemented with iron during their pregnancy (a highly iron demanding period), they are most likely to develop iron deficiency anemia [64].

The strengths of this study were; first, it was conducted using pooled data from 10 nationally representative DHS surveys in East African countries. Thus, this large sample size had adequate power to detect the true effect of the independent variables. Second, the sampling weight was applied during the analysis to get reliable estimates and standard errors. As a limitation, since the study used cross-sectional data, a causal relationship between anemia and the identified independent variables cannot be established. Besides, this study was based on secondary data, the factors that may be relevant to anemia during pregnancy such as eating habits, parasite infestations (hookworm), previous hospitalization, and use of nutritional supplements (vitamin B12 and folic acid) were not included.

## Conclusion

In this study, both individual and country-level factors were associated with anemia among pregnant women. Accordingly, unmarried women, teenagers, those women who had unproved toilet facility and women from a country with high illiteracy levels had a higher prevalence of anemia during pregnancy. Therefore, enabling the households to have improved

toilet facilities by strengthening the existing health extension program, improving the community literacy level, and minimizing teenage pregnancy is vital to reduce the prevalence of anemia in East Africa.

## Acknowledgments

The authors would like to thank measure DHS for their permission to access the EDHS datasets.

## Author Contributions

**Conceptualization:** Alemneh Mekuriaw Liyew, Getayeneh Antehunegn Tesema, Tesfa Sewunet Alamneh, Misganaw Gebrie Worku, Achamyeleh Birhanu Teshale, Adugnaw Zeleke Alem, Zemenu Tadesse Tessema, Yigizie Yeshaw.

**Data curation:** Alemneh Mekuriaw Liyew, Getayeneh Antehunegn Tesema, Tesfa Sewunet Alamneh, Misganaw Gebrie Worku, Achamyeleh Birhanu Teshale, Adugnaw Zeleke Alem, Zemenu Tadesse Tessema, Yigizie Yeshaw.

**Formal analysis:** Alemneh Mekuriaw Liyew, Getayeneh Antehunegn Tesema, Tesfa Sewunet Alamneh, Misganaw Gebrie Worku, Achamyeleh Birhanu Teshale, Adugnaw Zeleke Alem, Zemenu Tadesse Tessema, Yigizie Yeshaw.

**Investigation:** Alemneh Mekuriaw Liyew, Tesfa Sewunet Alamneh, Misganaw Gebrie Worku, Zemenu Tadesse Tessema, Yigizie Yeshaw.

**Methodology:** Alemneh Mekuriaw Liyew, Getayeneh Antehunegn Tesema, Tesfa Sewunet Alamneh, Misganaw Gebrie Worku, Achamyeleh Birhanu Teshale, Adugnaw Zeleke Alem, Zemenu Tadesse Tessema, Yigizie Yeshaw.

**Resources:** Alemneh Mekuriaw Liyew, Tesfa Sewunet Alamneh, Achamyeleh Birhanu Teshale, Adugnaw Zeleke Alem.

**Software:** Alemneh Mekuriaw Liyew, Getayeneh Antehunegn Tesema, Tesfa Sewunet Alamneh, Achamyeleh Birhanu Teshale, Adugnaw Zeleke Alem, Zemenu Tadesse Tessema, Yigizie Yeshaw.

**Supervision:** Alemneh Mekuriaw Liyew, Getayeneh Antehunegn Tesema.

**Validation:** Alemneh Mekuriaw Liyew, Getayeneh Antehunegn Tesema, Tesfa Sewunet Alamneh, Misganaw Gebrie Worku, Achamyeleh Birhanu Teshale, Adugnaw Zeleke Alem, Zemenu Tadesse Tessema, Yigizie Yeshaw.

**Visualization:** Alemneh Mekuriaw Liyew, Tesfa Sewunet Alamneh, Misganaw Gebrie Worku, Zemenu Tadesse Tessema, Yigizie Yeshaw.

**Writing – original draft:** Alemneh Mekuriaw Liyew.

**Writing – review & editing:** Alemneh Mekuriaw Liyew, Getayeneh Antehunegn Tesema, Tesfa Sewunet Alamneh, Misganaw Gebrie Worku, Achamyeleh Birhanu Teshale, Adugnaw Zeleke Alem, Zemenu Tadesse Tessema, Yigizie Yeshaw.

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
