## [Decision Letter · Decision Letter 0]

18 Feb 2021

PONE-D-20-37697

Prevalence and determinants of anemia among pregnant women in East Africa; A multilevel analysis of recent Demographic and Health Surveys

PLOS ONE

Dear Dr. Liyew,

Thank you for submitting your manuscript to PLOS ONE. After careful consideration, we feel that it has merit but does not fully meet PLOS ONE’s publication criteria as it currently stands. Therefore, we invite you to submit a revised version of the manuscript that addresses the points raised during the review process.

The study is relevant for planning interventions and actions in maternal and child health in the region. However, the reviewers have pointed out important topics for manuscript improvements. A major revision would be necessary addressing all reviewers´comments. Special attention should be considered to provide a more robust rationale and justification for this study. 

We look forward to receiving your revised manuscript.

Kind regards,

Marly A. Cardoso, Ph.D.

Academic Editor

PLOS ONE

Journal Requirements:

3. Please reference and discuss the following relevant publication:

https://www.ncbi.nlm.nih.gov/pmc/articles/PMC7485848/

Reviewers' comments:

Reviewer's Responses to Questions

**Comments to the Author**

1. Is the manuscript technically sound, and do the data support the conclusions?

Reviewer #1: Yes

Reviewer #2: Yes

2. Has the statistical analysis been performed appropriately and rigorously? 

Reviewer #1: I Don't Know

Reviewer #2: I Don't Know

3. Have the authors made all data underlying the findings in their manuscript fully available?

Reviewer #1: Yes

Reviewer #2: Yes

4. Is the manuscript presented in an intelligible fashion and written in standard English?

Reviewer #1: No

Reviewer #2: Yes

5. Review Comments to the Author

Reviewer #1: After had carefully read the manuscript and take notes on overall strengths and weaknesses, I share my comments below.

General comments

- This study has a considerable prevalence of anemia in pregnant women in East Africa. It also presents the determinants related to anemia in pregnant women. However, the results presented did not differ from other studies already published in this area, and some researches and analysis published in this journal are very similar to this article.

- There are some typos in the text that need to be carefully revised

- Broadly, it is not clear what this study adds to the broader literature on determinants of anemia among pregnant women.

- The authors need to provide a more robust justification or highlight why this study is necessary and adds more information to the previous studies.

- In the first paragraph, the authors introduce concepts about anemia and how it affects pregnancy. I recommend starting the introduction in the second paragraph to present the global problem of anemia and its prevalence in Southeast Asia and Africa. - Also, I suggest addressing more issues of how the article differs from articles already published on this topic (similar with women of reproductive age - https://doi.org/10.1371/journal.pone.0238957).

Results

- The readers would benefit if they had described the analysis plan (a figure or written) and how the analysis levels were included based on the previous literature and the novelty of this paper. Also, be more clear how which variables were chosen, as models control for covariates.

Reviewer #2: The manuscript addresses a relevant topic for improving the nutritional quality of pregnant women in East Africa. However, corrections and clarity are necessary in some issues, especially regarding the description of the method.

I made some comments that I hope will help you improve the manuscript.

Despite I am not a native of English it is important to review the written of your manuscript. I had some difficult to understand some phrases around your text.

1) Introduction

References are not always adequately used nor updated. The authors cite 39 references. Only 12 are from the last 5 years (2016-2020). One third of the references were published more than 10 years ago.

2) Methods

The study uses data from the Demographic and Health Surveys (DHS), which are nationally representative surveys and comparable across countries. However, I would suggest including a more detailed description of the DHS and the variables used. This reference may be useful: Daniel J Corsi, Melissa Neuman, Jocelyn E Finlay, SV Subramanian, Demographic and health surveys: a profile, International Journal of Epidemiology, Volume 41, Issue 6, December 2012, Pages 1602–1613, https://doi.org/10.1093/ije/dys184

It is important to highlight whether the pregnant women selected from DHS for the present study constitute a representative sample at the national level.

Dependent variable: the definition of anemia is confused. Why does the classification of anemia severity refer to women of reproductive age? “severe (if Hgb value <7 g / dL), and moderate (if Hgb value 7–9.9 g / dL) in women of reproductive age and mild (if Hgb level is 10.0–10.9 g/dL) in pregnant women. For this study, we recategorized anemia level as anemic which is coded as ‘1’ and non-anemic which is coded as ‘0’ from the previous classifications (no, mild, moderate, and severe).” Why did '0' include: no, mild, moderate, and severe?

Independent variables: it is necessary to describe each variable in more detail.

The order of presentation of the variables is very confusing. I would suggest that the authors review the order of presentation. Is it possible to present the broader social variables first? Example: 1) place of residence (urban, rural); 2) level of poverty (high, low); 3) level of illiteracy (high, low); 4) community health insurance (high, low).

AFTER presenting individual social, economic and demographic variables:

First, the broader social variables: 1-Wealth status (poorest, poor, middle, rich, richest); 2- Type of source of drinking water (improved, not improved); 3-type of toilet facility (improved, not improved); 4-Covered by health insurance (yes, no); 5-distance from the health facility (big problem, not a big problem); 5-media exposure (yes, no). I would suggest that the authors describe what these variables refer to, for example, media exposure…

Then, present the most proximal individual variables: 1-age (describe the age groups: 15-19, 20-24, 25-29, 30-34, 35-39, >= 40); 2-maternal education (no formal, primary, secondary, higher); 3-marital status (unmarried/married); 4-maternal working (yes, no); 5-sex of household head (male, female).

In the sequence, present the obstetric variables: 1-parity (….); 2-history of a terminated pregnancy (yes, no); 3-iron supplementation (yes, no).

3) Results

In Table 2, I would suggest present the variables in the same order as described in the methods. In addition, corrections are necessary: standardization of decimal places; in the wealth status variable, middle is out of place; in the age group it is need to correct 15-29 years and 34 years old is in two groups.

I would suggest to include the CIs of all countries in the Figure 1, and highlighting the values.

The item “Random effect and model comparison” presents the Tables 2, but wouldn't it be Table 3?

In Table 3, I would suggest to repeat the meaning of the acronyms in a footnote, and include the p-value.

In the item “Determinants of anemia among pregnant women”, the description of the results is confusing in relation to the individual and community variables. I would suggest to present the community variables first and then the individual ones, following the same order described in the method.

4) In the Introduction, the authors indicate that most factors associated with anemia during pregnancy are linked to socioeconomic conditions.

However, in the discussion, the comments do not refer to the worse social status of illiterate and unmarried women. For example, the authors justify that "This may be due to the low level of use of health services during pregnancy by illiterate women. There is evidence that these illiterate women did not seek prenatal care (ANC) (60)". Subsequently, the authors point out "In addition, illiterate women may be economically unstable (62) and may fail to meet nutritional needs (foods rich in vitamin A, iron and folic acid, etc.) during pregnancy, which can result in anemia”. However, to maintain consistency, this should be the first justification, as they do not have access to health services because they are socially excluded.

Here too, the references are not always used adequately and are not always up date.

Examples: end of 1sr paragraph: “Besides, the higher prevalence of anemia in the current study could be attributed to the recent resurgences of malaria in East Africa due to climate change (46)”. To support their statement, the authors cite a study of 2002, but they should cite more recent studies.

6. PLOS authors have the option to publish the peer review history of their article (what does this mean?). If published, this will include your full peer review and any attached files.

Reviewer #1: No

Reviewer #2: No

---

## [Author Response · Author response to Decision Letter 0]

10 Mar 2021

Rebuttal letter Date march /08/2021

Subject; submission of revised manuscript (PONE-D-20-37697)

Prevalence and determinants of anemia among pregnant women in East Africa; a multilevel analysis of recent Demographic and Health Surveys

Alemneh MekuriawLiyew

To PLOS ONE

Dear all,

We would like to thank you for these constructive, building, and improvable comments on this manuscript that would improve the substance and content of the manuscript. We considered each comment and clarification questions of editors and reviewers on the manuscript thoroughly. Our point-by-point responses for each comment and question are described in detail on the following pages. Further, the details of changes were shown by track changes in the supplementary document attached. The manuscript language was checked by language professionals and we follow journal guidelines. I have attached recent comments in a point-by-point response.

Version 1; editor’s comments

Authors’ response; thank you dear editor we have prepared the documents based on PLOS ONE requirement.

Authors’ response; thanks, dear editor. We have corrected the grammatical and spelling errors in the revised manuscript.

3. Please reference and discuss the following relevant publication:

https://www.ncbi.nlm.nih.gov/pmc/articles/PMC7485848/

Authors’ response; thank you dear editor for your commitment we have included the above reference. Please see the revised manuscript.

Version 2; reviewers’ comments

1. Is the manuscript technically sound, and do the data support the conclusions?

Reviewer #1: Yes

Reviewer #2: Yes

Authors’ response; thank you, dear reviewers

2 Have the authors made all data underlying the findings in their manuscript fully available?

The PLOS Data policy requires authors to make all data underlying the findings described in their manuscript fully available without restriction, with rare exceptions (please refer to the Data Availability Statement in the manuscript PDF file). The data should be provided as part of the manuscript or its supporting information, or deposited to a public repository. For example, in addition to summary statistics, the data points behind means, medians and variance measures should be available. If there are restrictions on publicly sharing data—e.g. participant privacy or use of data from a third party—those must be specified.

Reviewer #1: Yes

Reviewer #2: Yes

Authors’ response; thanks dear reviewers

3. Is the manuscript presented in an intelligible fashion and written in Standard English?

Reviewer #1: No

Reviewer #2: Yes

Authors’ response; thanks a lot dear reviewers we have critically improved the readability of the manuscript. Please see the revised version.

Reviewer 1 Comments 

After had carefully read the manuscript and take notes on overall strengths and weaknesses, I share my comments below.

General comments

comment 1; this study has a considerable prevalence of anemia in pregnant women in East Africa. It also presents the determinants related to anemia in pregnant women. However, the results presented did not differ from other studies already published in this area, and some researches and analyses published in this journal are very similar to this article.

- There are some typos in the text that need to be carefully revised

- Broadly, it is not clear what this study adds to the broader literature on determinants of anemia among pregnant women.

- The authors need to provide a more robust justification or highlight why this study is necessary and adds more information to the previous studies.

Authors’ response; thanks a lot dear reviewer for your critical comment to improve the manuscript. We accepted your comment and included sound justification in the revised manuscript. Though there are several publications for anemia during pregnancy so far, none of them give a summary or pooled prevalence of anemia and related determinants in east Africa. All are either country-based or sub-country-based studies. Nowadays integrating east Africa in different aspects including health is encouraged. Therefore, the findings from this study could have an important role to design maternal health programs in East Africa in an integrated approach. We have included a well-elaborated justification with clear citations in the revised manuscript. Please see paragraph 6 lines 79-93 of the revised manuscript.

Comment 2; In the first paragraph, the authors introduce concepts about anemia and how it affects pregnancy. I recommend starting the introduction in the second paragraph to present the global problem of anemia and its prevalence in Southeast Asia and Africa. - Also, I suggest addressing more issues of how the article differs from articles already published on this topic (similar with women of reproductive age - https://doi.org/10.1371/journal.pone.0238957).

Authors’ response; thanks a lot dear reviewer effort to improve the manuscript. As per your comment, the second paragraph describes the global burden of anemia during pregnancy followed by South East Asia Africa, sub-Saharan Africa, and East African countries. We also justified the benefit of the current study over the above-given article. We kindly request to see the revised manuscript.

Results

Comment 3; the readers would benefit if they had described the analysis plan (a figure or written) and how the analysis levels were included based on the previous literature and the novelty of this paper. Also, be clearer how variables were chosen, as models control for covariates.

Authors’ response; thanks, dear reviewer. The analysis was not based on the traditional regression methods. We used the advanced model with the multilevel analysis approached. Therefore, as per your comment, we clearly described the type of model used (multilevel binary logistic regression) level one and level two variables.

Reviewer #2 comments

Comment and suggestion; The manuscript addresses a relevant topic for improving the nutritional quality of pregnant women in East Africa. However, corrections and clarity are necessary for some issues, especially regarding the description of the method.

I made some comments that I hope will help you improve the manuscript.

Despite I am not a native of English it is important to review the writing of your manuscript. I had some difficulty understanding some phrases around your text.

Authors’ response; thanks dear reviewer we accepted your comment and the readability of the revised manuscript is well improved.

1) Introduction

comment 2; References are not always adequately used nor updated. The authors cite 39 references. Only 12 are from the last 5 years (2016-2020). One-third of the references were published more than 10 years ago.

Authors’ response; thank your dear reviewer for your strong effort to improve the quality of our manuscript and to keep the scientific rigor. We tried to search for the most recent publications. While doing so we replaced some old references with the updated ones. However, we are forced to use some old references due to lack of the updated articles.

2) Methods

comment 3; The study uses data from the Demographic and Health Surveys (DHS), which are nationally representative surveys and comparable across countries. However, I would suggest including a more detailed description of the DHS and the variables used. This reference may be useful: Daniel J Corsi, Melissa Neuman, Jocelyn E Finlay, SV Subramanian, Demographic and health surveys: a profile, International Journal of Epidemiology, Volume 41, Issue 6, December 2012, Pages 1602–1613, https://doi.org/10.1093/ije/dys184

It is important to highlight whether the pregnant women selected from DHS for the present study constitute a representative sample at the national level.

Authors’ response; thank you dear reviewer we strongly appreciate your commitment. We learned a lot from “Daniel J Corsi, Melissa Neuman, Jocelyn E Finlay, SV Subramanian, Demographic and health surveys: a profile, International Journal of Epidemiology, Volume 41, Issue 6, December 2012, Pages 1602–1613, https://doi.org/10.1093/ije/dys184” article. Consequently, we provided a detailed description of DHS data. Please see the revised manuscript.

Comment 4; Dependent variable: the definition of anemia is confusing. Why does the classification of anemia severity refer to women of reproductive age? “severe (if Hgb value <7 g / dL), and moderate (if Hgb value 7–9.9 g / dL) in women of reproductive age and mild (if Hgb level is 10.0–10.9 g/dL) in pregnant women. For this study, we recategorized anemia level as anemic which is coded as ‘1’ and non-anemic which is coded as ‘0’ from the previous classifications (no, mild, moderate, and severe).” Why did '0' include: no, mild, moderate, and severe?

Authors’ response; thank you, dear reviewer. Sorry for creating confusion. It is the grammatical error in the sentence construction “0” which includes only nonanemic ones. All other categories such as mild, moderate and severe are coded as “1’’. Generally, we paraphrased the whole paragraph in this section to improve the clarity. 

Comment 5; Independent variables: it is necessary to describe each variable in more detail.

The order of presentation of the variables is very confusing. I would suggest that the authors review the order of the presentation. Is it possible to present the broader social variables first? Example: 1) place of residence (urban, rural); 2) level of poverty (high, low); 3) level of illiteracy (high, low);4)community health insurance (high, low). AFTER presenting individual social, economic, and demographic variables: First, the broader social variables: 1-Wealth status (poorest, poor, middle, rich, richest); 2- Type of source of drinking water (improved, not improved); 3-type of toilet facility (improved, not improved); 4-Covered by health insurance (yes, no); 5-distance from the health facility (big problem, not a big problem); 5-media exposure (yes, no). I would suggest that the authors describe what these variables refer to, for example, media exposure… Then, present the most proximal individual variables: 1-age (describe the age groups: 15-19, 20-24, 25-29, 30-34, 35-39, >= 40); 2-maternal education (no formal, primary, secondary, higher); 3-marital status (unmarried/married); 4-maternal working (yes, no); 5-sex of household head (male, female).In the sequence, present the obstetric variables: 1-parity (….); 2-history of a terminated pregnancy (yes, no); 3-iron supplementation (yes, no).

Authors’ response; Thanks a lot dear reviewer for your interesting and constructive comments. We described each variable in detail by providing how it is categorized. Regarding the order of presentation of independent variables as you said, it is possible to present the variables from broader ones to most proximal ones in traditional regression analysis. However, in multi-level analysis, the conceptualization and presentation of variables should indicate the analytic levels. For example, in our case, we do have two level variables i.e individual level (level one) and community (country) level (level two). So in our method, we first presented all variables that operate at an individual level, and next, we presented level two variables. We also follow the same approach in the result section again. First, individual-level variables were described.

3) Results

comment 6; In Table 2, I would suggest presenting the variables in the same order as described in the methods. Besides, corrections are necessary: standardization of decimal places; in the wealth status variable, the middle is out of place; in the age group it needs to correct 15-29 years and 34 years old in two groups.

Authors’ response; thanks dear reviewer we appreciate your critics reading. We corrected all the issues accordingly.

Comment 7; I would suggest including the CIs of all countries in Figure 1, and highlighting the values.

The item “Random effect and model comparison” presents Tables 2, but wouldn't it be Table 3? In Table 3, I would suggest repeating the meaning of the acronyms in a footnote, and include the p-value.

Authors’ response; thanks a lot dear reviewer. Sorry for creating the inconvenience it is to say “table 3”. Now it is corrected acronyms were expanded at the footnote and the p-value was included as per your suggestion. The CIs were also included in fig 1.

Comment 8; In the item “Determinants of anemia among pregnant women”, the description of the results is confusing concerning the individual and community variables. I would suggest presenting the community variables first and then the individual ones, following the same order described in the method.

Authors’ response; thank you, dear reviewer. As we have described in the above response for “comment 5”. In the same way with that of the method section here also we first described individual-level variables and then the community-level ones.

Comment 9; In the Introduction, the authors indicate that most factors associated with anemia during pregnancy are linked to socioeconomic conditions.

However, in the discussion, the comments do not refer to the worse social status of illiterate and unmarried women. For example, the authors justify that "This may be due to the low level of use of health services during pregnancy by illiterate women. There is evidence that these illiterate women did not seek prenatal care (ANC) (60)". Subsequently, the authors point out "In addition, illiterate women may be economically unstable (62) and may fail to meet nutritional needs (foods rich in vitamin A, iron and folic acid, etc.) during pregnancy, which can result in anemia”. However, to maintain consistency, this should be the first justification, as they do not have access to health services because they are socially excluded.

Authors’ response; thanks, reviewer. We accepted your comment and corrected it accordingly. We kindly request you to see the revised version.

Comment 10; Here too, the references are not always used adequately and are not always update. Examples: end of 1sr paragraph: “Besides, the higher prevalence of anemia in the current study could be attributed to the recent resurgences of malaria in East Africa due to climate change (46)”. To support their statement, the authors cite a study of 2002, but they should cite more recent studies.

Authors’ response; thanks dear reviewer for your suggestion we replaced the above reference with relatives recent publication. But we face a challenge to get updated references for most of the articles. Consequently, we are forced to use the existing and available kinds of literature.

Thanks, a lot!!!

---

## [Decision Letter · Decision Letter 1]

26 Mar 2021

PONE-D-20-37697R1

Prevalence and determinants of anemia among pregnant women in East Africa; A multilevel analysis of recent Demographic and Health Surveys

PLOS ONE

Dear Dr. Liyew,

Thank you for submitting your manuscript to PLOS ONE. After careful consideration, we feel that it has merit but does not fully meet PLOS ONE’s publication criteria as it currently stands. Therefore, we invite you to submit a revised version of the manuscript that addresses the points raised during the review process.

The authors have addressed almost all reviewers´ comments. However, the manuscript still needs revision following the reviewer suggestions, with special attention to the description of the changes in prevalence of anemia among countries when compared with previous version (and why). 

We look forward to receiving your revised manuscript.

Kind regards,

Marly A. Cardoso, Ph.D.

Academic Editor

PLOS ONE

Journal Requirements:

Reviewers' comments:

Reviewer's Responses to Questions

**Comments to the Author**

1. If the authors have adequately addressed your comments raised in a previous round of review and you feel that this manuscript is now acceptable for publication, you may indicate that here to bypass the “Comments to the Author” section, enter your conflict of interest statement in the “Confidential to Editor” section, and submit your "Accept" recommendation.

Reviewer #2: (No Response)

2. Is the manuscript technically sound, and do the data support the conclusions?

Reviewer #2: Partly

3. Has the statistical analysis been performed appropriately and rigorously? 

Reviewer #2: I Don't Know

4. Have the authors made all data underlying the findings in their manuscript fully available?

Reviewer #2: Yes

5. Is the manuscript presented in an intelligible fashion and written in standard English?

Reviewer #2: Yes

6. Review Comments to the Author

Reviewer #2: The authors reviewed almost all the items highlighted, but there is a major problem, which refers to changes in the prevalence of anemia, without justification.

There was a small change in the overall prevalence of anemia, but a substantial change in the prevalence of anemia in each of the countries, as shown in Figure 1:

1st version:

The prevalence of anemia among pregnant women in eastern Africa was 42.50 (95% CI: 41.52, 44.10) with a large difference between specific countries which ranges from 22.08% in Burundi to 51.88% in Zimbabwe (Fig. 1).

Figure 1: Burundi (22.08), Ethiopia (32.45), Madagascar (35.20), Malawi (37.32), Mozambique (38.02), Rwanda (38.64), Tanzania (44.94) ), Uganda (45.48), Zambia (51.88), Zimbabwe (58.99)

2nd version:

The prevalence of anemia among pregnant women in easten Africa was 41.82 (95% CI: 40.78, 42.87) with a large difference between specific countries which range from 23.36% in Rwanda to 57.10% in Tanzania (Fig. 1).

Figure 1: Burundi (46.76), Ethiopia (28.01), Madagascar (37.44), Malawi (45.12), Mozambique (50.64), Rwanda (23.38), Tanzania (57.10) ), Uganda (38.20), Zambia (41.19), Zimbabwe (33.07)

Despite these substantial changes in prevalence, the other analyzes have not changed and I have found no justification for these changes in the authors' response.

Other minor adjustments are necessary for the manuscript to be accepted for publication.

In Table 2:

a) I suggest including a line identifying the variables at the Individual-level and another line indicating the variables at the Community-level.

b) I continue to suggest maintaining the same order as the variables presented in the Method.

c) It is necessary to correct the maternal age of 15-29 years group.

In Table 3:

a) In the variable Maternal occupation, in Model II, it is necessary to correct 1.2 (0.93,1.11). Would it be 1.02?

b) In the variable Maternal age: In Model III, what was the reference? Should not the asterisk in the 20 to 24 age group be in the 15 to 19 age group?

7. PLOS authors have the option to publish the peer review history of their article (what does this mean?). If published, this will include your full peer review and any attached files.

Reviewer #2: No

---

## [Author Response · Author response to Decision Letter 1]

27 Mar 2021

Rebuttal letter Date march /27/2021

Subject; submission of revised manuscript (PONE-D-20-37697R1)

Prevalence and determinants of anemia among pregnant women in East Africa; a multilevel analysis of recent Demographic and Health Surveys

Alemneh Mekuriaw Liyew

To PLOS ONE

Dear all,

We would like to thank you for these constructive, building, and improvable comments on this manuscript that would improve the substance and content of the manuscript. We considered each comment and clarification questions of editors and reviewers on the manuscript thoroughly. Our point-by-point responses for each comment and question are described in detail on the following pages. Further, the details of changes were shown by track changes in the supplementary document attached. We have attached recent comments in a point-by-point response.

Version 1; editor’s comments

Authors’ response; thank you dear editor we have prepared the documents based on PLOS ONE requirement.

Authors’ response; thanks dear editor we can confirm that our reference is complete and correcte. Retracted paper is not cited.

Version 2; reviewer’s comments

1. Is the manuscript technically sound, and do the data support the conclusions?

Reviewer #1: Yes

Reviewer #2: Yes

Authors’ response; thank you, dear reviewers

2 Have the authors made all data underlying the findings in their manuscript fully available?

The PLOS Data policy requires authors to make all data underlying the findings described in their manuscript fully available without restriction, with rare exceptions (please refer to the Data Availability Statement in the manuscript PDF file). The data should be provided as part of the manuscript or its supporting information, or deposited to a public repository. For example, in addition to summary statistics, the data points behind means, medians and variance measures should be available. If there are restrictions on publicly sharing data e.g. participant privacy or use of data from a third party—those must be specified.

Reviewer #1: Yes

Reviewer #2: Yes

Authors’ response; thanks dear reviewer

Reviewer Comments 

Comment 1; The authors reviewed almost all the items highlighted, but there is a major problem, which refers to changes in the prevalence of anemia, without justification. There was a small change in the overall prevalence of anemia, but a substantial change in the prevalence of anemia in each of the countries, as shown in Figure 1:

1st version:

The prevalence of anemia among pregnant women in eastern Africa was 42.50 (95% CI: 41.52, 44.10) with a large difference between specific countries which ranges from 22.08% in Burundi to 51.88% in Zimbabwe (Fig. 1).

Figure 1: Burundi (22.08), Ethiopia (32.45), Madagascar (35.20), Malawi (37.32), Mozambique (38.02), Rwanda (38.64), Tanzania (44.94) ), Uganda (45.48), Zambia (51.88), Zimbabwe (58.99)

2nd version:

The prevalence of anemia among pregnant women in easten Africa was 41.82 (95% CI: 40.78, 42.87) with a large difference between specific countries which range from 23.36% in Rwanda to 57.10% in Tanzania (Fig. 1).

Figure 1: Burundi (46.76), Ethiopia (28.01), Madagascar (37.44), Malawi (45.12), Mozambique (50.64), Rwanda (23.38), Tanzania (57.10) ), Uganda (38.20), Zambia (41.19), Zimbabwe (33.07)

Despite these substantial changes in prevalence, the other analyzes have not changed and I have found no justification for these changes in the authors' response.

Authors’ response; thanks dear reviewer for your critical comment. We ask a great apology for creating such inconvenance and unnecessary confussion. As you know in the large survey data like DHS (the one we used ) there is a great difference in the weighted and unweighted prevalence. The weighted prevalence provides better and valid eastimate than the unweighted one . Thus,it is the weighted prevalence that we are expected to report. (Guide to DHS Statistic 2018). However, in the first version the unweighted proportion was reported though the sampling weight was adjusted in the regression analysis. 

Therefore, in the revised version the weighted prevalence was conducted for each country (that is why it is completely different from the first version) as well as for the overall prevalence by using the weighting variable which is already available in in the datasets. So, the updated prevalence (second version) is valid estimate than the previous one for two reasons 

1. The sampling weight was applied to produce the actual and reliable estimate

2. The confidence interval was included for those weighted prevalences

Regarding the regression analysis, the sampling weight was already adjusted in stata in all models to produce reliable estimates. That is why the findings in the first version and in the second version for the regression outputs are similar. As you know prevalence (descriptive only; for which we missed the sampling weight in the previous version) and regression analysis ( for which we accounted the sampling weight in the first version) are conducted separately. In describing the frequency for each independent variables we have already reported both the unweighted and weighted frequencies with respective percentage in both versions for clarity.

So, the mistake is that we fail to justify such happening in our previous point by point response. We ask a great appology for this.

comment 2. In Table 2:

a) I suggest including a line identifying the variables at the Individual-level and another line indicating the variables at the Community-level.

b) I continue to suggest maintaining the same order as the variables presented in the Method.

c) It is necessary to correct the maternal age of 15-29 years group.

Authors’ response; thanks dear reviewer. We have fully accepted three of your comments and corrected accordingly.

comment 3 ;In Table 3:

a) In the variable Maternal occupation, in Model II, it is necessary to correct 1.2 (0.93,1.11). Would it be 1.02?

b) In the variable Maternal age: In Model III, what was the reference? Should not the asterisk in the 20 to 24 age group be in the 15 to 19 age group?

Authors’ response; we really appreciate your critical reading ability. We have corrected both of your comments accordingly.

Thanks, a lot!!!

---

## [Editor Report · Decision Letter 2]

12 Apr 2021

Prevalence and determinants of anemia among pregnant women in East Africa; A multilevel analysis of recent Demographic and Health Surveys

PONE-D-20-37697R2

Dear Dr. Liyew,

We’re pleased to inform you that your manuscript has been judged scientifically suitable for publication and will be formally accepted for publication once it meets all outstanding technical requirements.

Kind regards,

Marly A. Cardoso, Ph.D.

Academic Editor

PLOS ONE
---

## [Editor Report · Acceptance letter]

16 Apr 2021

PONE-D-20-37697R2 

Prevalence and determinants of anemia among pregnant women in East Africa;A multi level analysis of recent Demographic and Health Surveys 

Dear Dr. Liyew:

I'm pleased to inform you that your manuscript has been deemed suitable for publication in PLOS ONE. Congratulations! Your manuscript is now with our production department. 

Kind regards, 

on behalf of

Dr. Marly A. Cardoso 

Academic Editor

PLOS ONE